# Comprehensive Comparison of Two Fault Tolerant Axial Field Modular Flux-Switching Permanent Magnet Machines with Different Stator and Rotor Pole-Pairs Combinations

**Yixiang Tu [1], Mingyao Lin [1,\*], Keman Lin [2], Yong Kong [1] and Da Xu [3]**

[1] School of Electrical Engineering, Southeast University, Nanjing 210096, China; tuyixiang@seu.edu.cn (Y.T.); kongyong@seu.edu.cn (Y.K.)
[2] College of Energy and Electrical Engineering, Hohai University, Nanjing 211100, China; linkeman@hhu.edu.cn
[3] School of Automation, Nanjing University of Science and Technology, Nanjing 210014, China; xuda@njust.edu.cn
\* Correspondence: mylin@seu.edu.cn

**Abstract:** This paper gives a comprehensive comparison among two fault-tolerant axial field modular flux-switching (AFFSPM) machines with different stator modular segments (U- and E-core) and stator-slots/pole-pairs combinations. The topologies of two AFFSPM machines are introduced, each with two feasible stator slots and rotor pole-pairs combinations with high winding factors based on the slot-conductor back-EMF star vectors theory. Then, the static performance including the air-gap flux density, flux linkage, back-electromagnetic force (back-EMF), and electromagnetic torque are analyzed and compared. Moreover, the fault-tolerant capability is then investigated by the torque performance under one- and two-phase open-circuit conditions in which the corresponding fault-tolerant control strategies are applied. The predicted results confirm that the 6-stator slot/11-rotor-pole-pair E-core AFFSPM machine exhibits the best performance of the four candidates.

**Keywords:** axial field flux-switching permanent magnet (AFFSPM) machine; winding factor; fault tolerant; rotating field reconstruction

## 1. Introduction

In recent decades, electrical vehicles (EVs) have become an effective solution for the energy shortage crisis and decreasing environmental pollution [1]. Traditional interior permanent magnet (IPM) machines have been widely used in EVs, like the BMW i3, Toyota Prius, etc. [2–4]. However, the PMs in these vehicles are located at the rotor, where the intense heat generated by eddy currents is difficult to dissipate. Furthermore, the long end winding of distributed winding configuration leads to extra copper loss. Both eddy current loss and copper loss increase the risk of the PM demagnetization. In order to overcome the potential defections above, a stator-permanent magnet machine called the flux-switching permanent magnet (FSPM) machine is proposed, in which the PMs are moved to the stator from the rotor [5]. Due to their high efficiency and robust structure, FSPM machines have attracted considerable attention, and a great deal of research has been implemented [6–8]. In these models, the PM is located at the stator yoke, which easily dissipates the heat [9]. However, this positioning will also lead to a unipolar PM flux linkage and lower back-electromotive force (back-EMF). In this paper, a 12 stator slots/10 rotor pole-pairs (12s/10p) FSPM machine with different winding configurations is analyzed to obtain a bipolar EMF and larger torque [10,11]. The axial field flux-switching permanent magnet (AFFSPM) machine is also proposed to make axial length shorter and direct a better torque density and power density. The AFFSPM machine can be grouped by different stator topologies, i.e., U-core and E-core stators segments modules. A novel dual rotor U-core AFFSPM machine is investigated, and the cogging torque is optimized by various shapes of stator and rotor teeth [12–14]. A fault-tolerant U-core AFFSPM machine with different stator and

rotor combinations are analyzed to achieve a high winding factor and less eddy current loss [15]. A hybrid-excited U-core stator axial field flux-switching permanent magnet machine is also investigated, in which the air-gap field can be easily regulated, and the mutual inductance is largely reduced [16,17]. Another novel hybrid excitation 6s/10p E-core AFFSPM machine is studied to realize a wide speed range by flux regulation [18] and a harmonic field current injection method, and three novel topologies are further investigated for cogging torque reduction [19]. A 3-D magnetic equivalent circuit network that integrates the hysteresis model of an axial field flux-switching memory machine is presented to offer reasonable accuracy with moderate computational effort [20]. Several topologies of a 6s/4p C-core AFFSPM machine with a shifting angle between the stator and rotor are analyzed for the elimination of second-harmonic component [21]. A comparison between one stator permanent magnet flux-switching machine and one rotor permanent magnet flux-switching machine conducted from electromagnetic torque production mechanism and torque-sizing equation is given in [22]. A multi-tooth FSPM with a pair of magnets located in the upper apex of the stator tooth on the basis of the conventional 12s/17p FSPM machine is proposed, and four different rotor teeth combinations are compared and the results reveal that 6s/17p is the optimal combination [23]. Additionally, an axial field flux-switching permanent magnet machine and a radial field flux-switching permanent magnet machine are compared under the same conditions, and the results indicate that the axial field machine has greater torque in the constant power range [24]. It is important to note that this investigation is mainly limited by the design and optimization of radial and axial field FSPM machine with novel structures, as the comparison between different AFFSPM machines is rarely reported.

It is worth mentioning that the two sets of armature windings in AFFSPM machines located at both sides of the stator provide a natural physical isolation. This helps a novel double three-phase armature winding configuration to be derived, which is helpful for the fault-tolerant operation and gives a better torque performance than the radial field machine. However, the comparison of the fault-tolerant capability of AFFSPM machines has not been studied yet. Therefore, a comparative investigation between U- and E-core AFFSPM machines with different stator slots and rotor pole-pairs combinations will be given in detail in this paper.

Through the investigation and comparison of the AFFSPM machines with various topologies, an effective general design guide regarding the stator slots/rotor pole-pairs combinations, static characteristics and fault-tolerant capability can be obtained. In Section 2, the topologies of AFFSPM machines are introduced, and the feasible candidates with higher winding factor is confirmed by corresponding stator slots/rotor pole-pairs combinations principles in Section 3. In Section 4, the static characteristics, i.e., air gap flux distribution, PM flux linkage, back-EMF, winding inductance, and torque performance are analyzed and compared. In Section 5, the fault-tolerant capability of the machines is evaluated through the torque performance under one- and two-phase open-circuit conditions. In Section 6, some conclusions of the investigation are summarized.

## 2. Topologies of AFFSPM Machines

The topology of AFFSPM machines with U- and E-core stator segments are presented in Figure 1, respectively. The rotor structure is only laminated by iron sheets. The stator is constituted of stator segments that are sandwiched by circumferentially magnetized PMs with alternate polarity, which is given in Figure 2. Contrarily to the U-core segments, the middle-tooth of the E-core segment reduces the number of PMs and stator segments by half that of the U-core AFFSPM machine. As Figure 3 shows, the open winding configuration is applied to the machines and each phase is supplied with a H-bridge converter and isolated from the others, allowing high fault-tolerant capability to be achieved. The armature windings are wound across the PM and adjacent stator teeth. Therefore, the single layer concentrated winding is arranged in the E-core AFFSPM machine, and the U-core AFFSPM machine adopts the double layers concentrated windings. For fair comparison, the key

geometric parameters, including stator outside diameter, stack length, airgap are kept same, and the material of PM and iron cores are also identical, as listed in Table 1. It should be noted that the key dimensions of both AFFSPM machines have already been multi-objective optimized in the previous investigation to ensure that they carry out the best performance under the conditions in this paper. The AFFSPM machine can work in both generator and motor two modes. From a "generator-oriented" perspective, the PM flux linkage of the individual coil changes periodically as the rotor rotates continually, which can be explained by flux-switching theory [10,12]. Moreover, magnetic field modulation principle is reiterated from a motor-oriented perspective [24,25], where it can be found that the torque is mainly contributed by the dominant harmonics in PM field and armature reaction field with the same order and rotating speed.

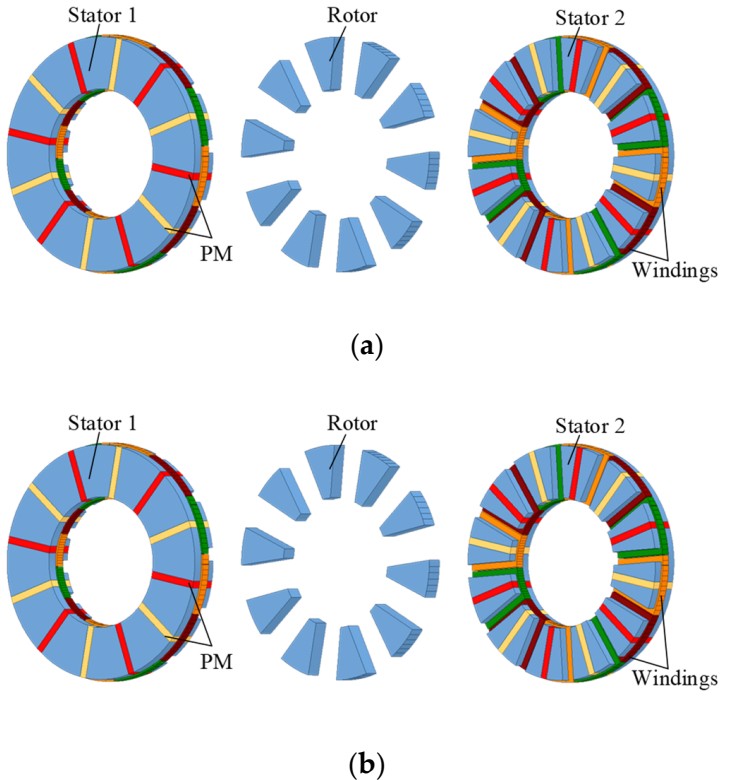

**Figure 1.** Topology of two AFFSPM machines. (**a**) AFFSPM machine with U-core stator segments; (**b**) AFFSPM machine with E-core stator segments.

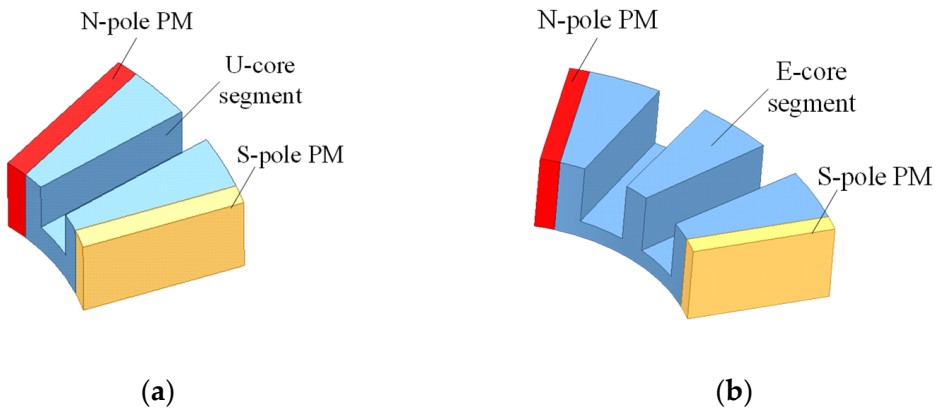

**Figure 2.** Configuration of two different modular stator segments. (**a**) U-core stator segments and PMs; (**b**) E-core stator segments and PMs.

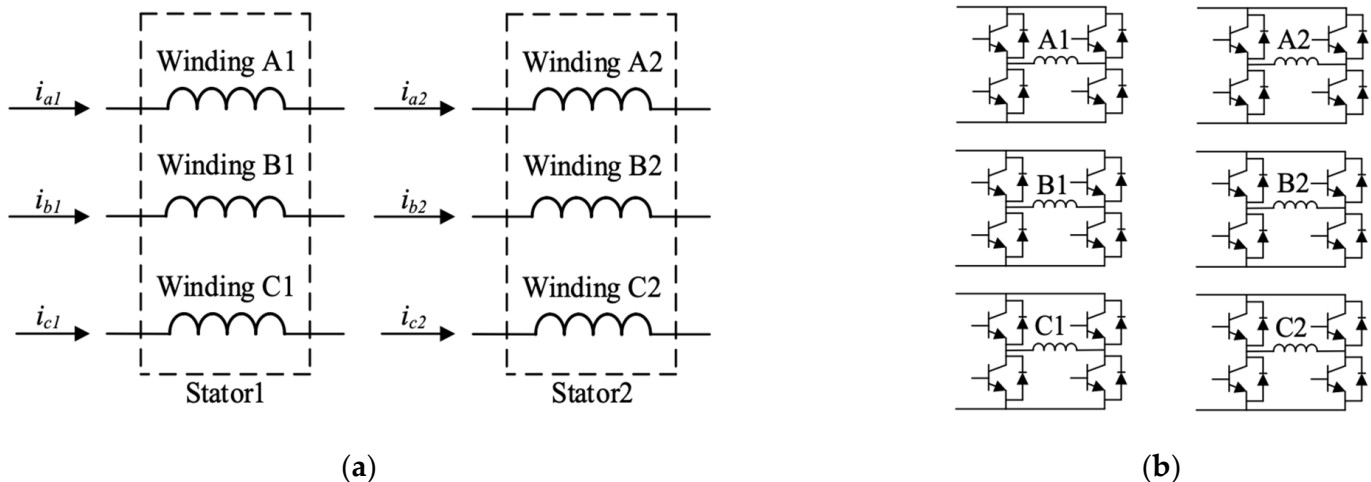

**Figure 3.** Schematic diagram of double three phase open winding and H-bridge inverter drive topology. (**a**) Double three open winding configuration; (**b**) H-bridge inverter drive topology.

**Table 1.** Key specifications and dimensions of two axial field flux-switching machines.

| Specifications | U-Core AFFSPM Machine | E-Core AFFSPM Machine |
|---|---|---|
| Based speed $n_b$ (rpm) | | 1500 |
| Axial length $h_a$ (mm) | | 67 |
| Air gap length $g$ (mm) | | 1 |
| Outer diameter $D_o$ (mm) | | 198.4 |
| Inner diameter $D_i$ (mm) | 109.1 | 119 |
| Stator teeth arc $\beta_{st}$ (deg.) | 9 | 7.5 |
| Stator middle teeth $\beta_{smt}$ (deg.) | 15 | – |
| Rotor teeth arc $\beta_{rt}$ (deg.) | 14.5 | 12.5 |
| Stator slot arc $\beta_{rs}$ (deg.) | 7.25 | 7.5 |
| PM width $\beta_{PM}$ (deg.) | 12.5 | 7.55 |
| PM height $h_{PM}$ (mm) | 25.5 | 21 |
| PM type | | N38SH |
| Iron lamination type | | 50W310 |

## 3. Stator Slots and Rotor Pole-Pairs Combination Principles

The different stator modules result in various stator slots and rotor pole-pairs combination principles of two machines. The stator slots number $P_s$ and rotor pole-pairs number $P_r$ of U- and E-core AFFSPM machines satisfies the Equations (1) and (2), respectively [11,26]. In order to reduce the iron loss of the machine and switch loss of the controllers, $P_r$ is usually less than $P_s$ in U-core AFFSPM machines, and less than $2P_s$ in E-core AFFSPM machines. The winding factor ($k_w$) of each combination can be calculated based on the slot-conductor back-EMF star vectors theory. Therefore, there are four feasible candidates (12/10 U-core, 12/11 U-core, 6/10 E-core, 6/11 E-core) with higher $k_w$, as outlined in Table 2. However, the performance of the AFFSPM machine can not only be confirmed by winding factor, and some static performance should also be analyzed for the selected combinations of $P_s/P_r$.

$$\begin{cases} P_s = mN_c \\ P_r = P_s \pm k \end{cases} \tag{1}$$

$$\begin{cases} P_s = mN_c \\ P_r = 2P_s \pm k \end{cases} \tag{2}$$

where $m$ is the phase number; $N_c$ is the coils number of the one-phase winding; $k = 1, 2, \ldots$

**Table 2.** Combination of $P_s$ and $P_r$ for U- and E-core AFFSPM machine.

| Item | $m$ | $P_s$ | $P_r$ | $k_w$ |
|---|---|---|---|---|
| U-core AFFSPM machine | 3 | 12 | 10 | 0.866 |
| | | 12 | 11 | 0.966 |
| E-core AFFSPM machine | 3 | 6 | 10 | 0.866 |
| | | 6 | 11 | 0.966 |

## 4. Static Performance Evaluation

### 4.1. Air-Gap Flux Density

According to the field modulation principle, the stator-side PMs provide a stationary permanent magnet magnetomotive force (PM-MMF) modulated by the magnetic permeance model of the salient rotor poles. Furthermore, the stationary MMF can be modulated into an air-gap magnetic field composed of abundant harmonics. The open-circuit air-gap flux density and the flux density harmonic distributions of four feasible combinations are shown in Figure 4. In addition, the curves of air-gap flux density and the harmonic distribution of four candidates are obtained by finite elements analysis (FEA) methods. It can be seen that the peak value of air-gap flux density of U-core AFFSPM machine is a little higher than that of the E-core AFFSPM machine due to the stronger flux-focusing effect. The flux density harmonic distribution is obtained through the fast Fourier transform (FFT) analysis in MATLAB. For both U- and E-core machines, it can be concluded that the effective harmonics can be classified into two types; the dominant harmonic components are produced by the primitive PM-MMF ($iP_{PM}$, $i = 1, 2, \dots$) and the other harmonics are generated by the modulated PM-MMF ($|iP_{PM} \pm jP_r|$, $i, j = 1, 2, \dots$), which is summarized in Table 3.

**Table 3.** Characteristics of no-load air-gap flux-density harmonics of the four candidates.

| Item | Dominant Harmonics ($iP_{PM}$) | Modulated Harmonics ($|iP_{PM} \pm jP_r|$) |
|---|---|---|
| 12/10 U-core | 6 ($i = 1$), 18 ($i = 3$) | 4, 16 ($i = 1, j = 1$)<br>8, 28 ($i = 3, j = 1$)<br>14, 34 ($i = 4, j = 1$) |
| 12/11 U-core | 6 ($i = 1$), 18 ($i = 3$) | 5, 17 ($i = 1, j = 1$)<br>7, 29 ($i = 3, j = 1$) |
| 6/10 E-core | 3 ($i = 1$), 9 ($i = 3$), 15 ($i = 5$) | 7, 13 ($i = 1, j = 1$)<br>5, 25 ($i = 5, j = 1$) |
| 6/11 E-core | 3 ($i = 1$), 9 ($i = 3$), 15 ($i = 5$) | 8, 14 ($i = 1, j = 1$)<br>4, 26 ($i = 5, j = 1$) |

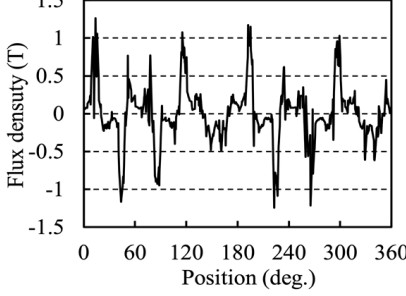 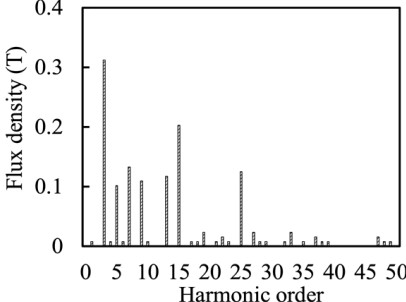

(**a**)

**Figure 4.** *Cont*.

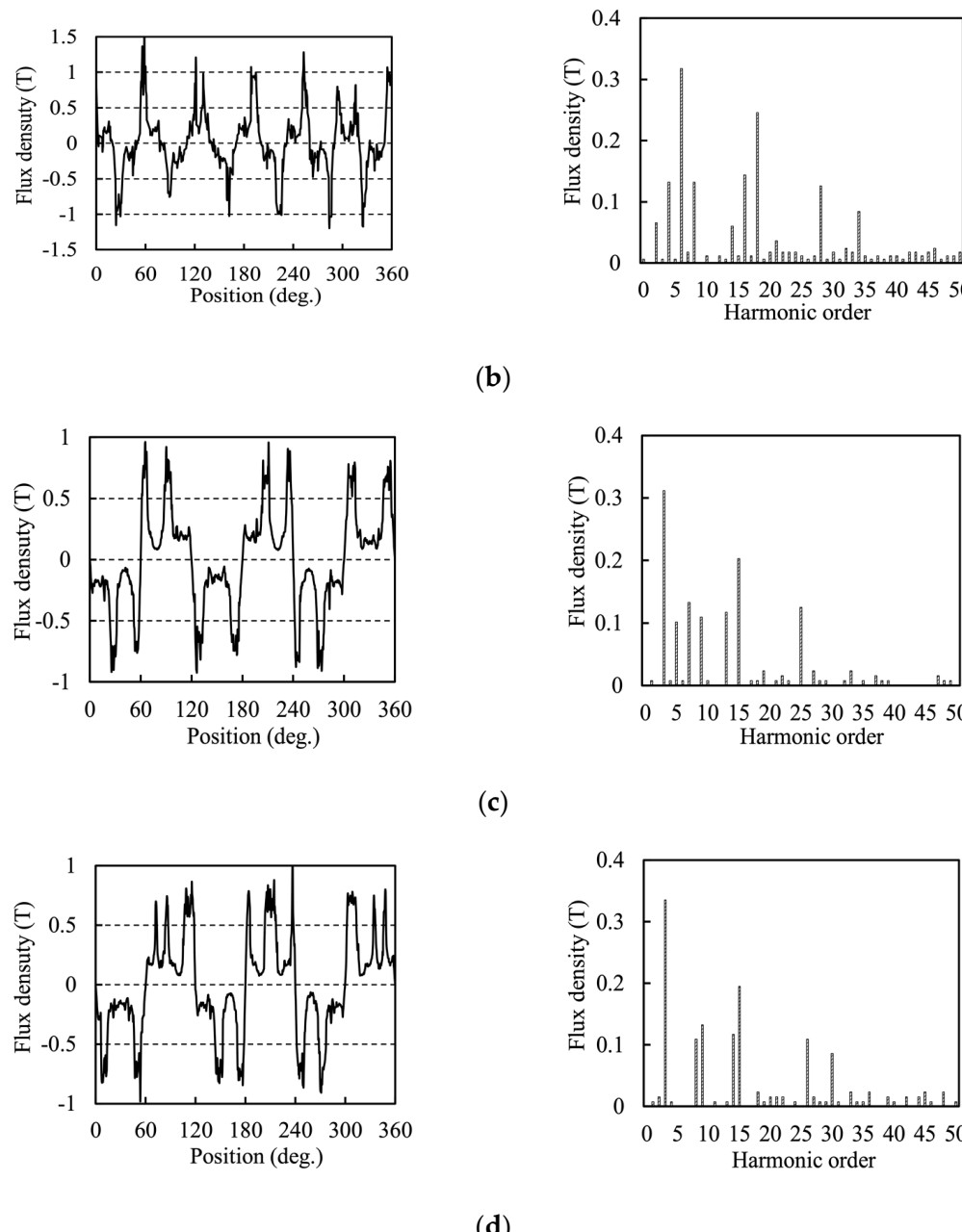

**Figure 4.** Open-circuit air-gap flux density and harmonic distribution of the four candidates. (**a**) 12/10 U-core; (**b**) 12/11 U-core; (**c**) 6/10 E-core; (**d**) 6/11 E-core.

*4.2. PM Flux Linkage and Back-EMF*

The no-load phase PM flux linkage and harmonic distribution of the four candidates are given in Figure 5. The peak value of the no-load phase PM flux linkage of the U-core AFFSPM machine is slightly larger than that of the E-core AFFSPM machine because of the larger air-gap flux density, as indicated in Section 4.1. The U-core AFFSPM machine contains higher odd order PM flux-linkage harmonics, while the E-core machine has a larger amount of even numbers. It is worth noting that, for 12/10 U-core and 6/10 E-core AFFSPM machines, there is no phase shifting in flux-linkage waveform among coils belonging to the same phase. However, there is a half period of phase angle shifting for 12/11 U-core and 6/11 E-core machines, which makes harmonic component offset by the combination of the PM flux of the coil in one phase. Therefore, the PM flux linkage of the 6/11 E-core and

12/11 U-core machines shows lower total harmonic distortion (THD) than the other two candidates due to the winding complementarity of AFFSPM machine.

Correspondingly, the amplitude of the fundamental phase back-EMF can be obtained and the open-circuit phase back-EMF waveforms at 3000 rpm are shown in Figure 6. By applying the FFT analysis, it can be found that there are the higher harmonic components in the 10-rotor pole-pair machines, which is in agreement with the PM flux-linkage analysis.

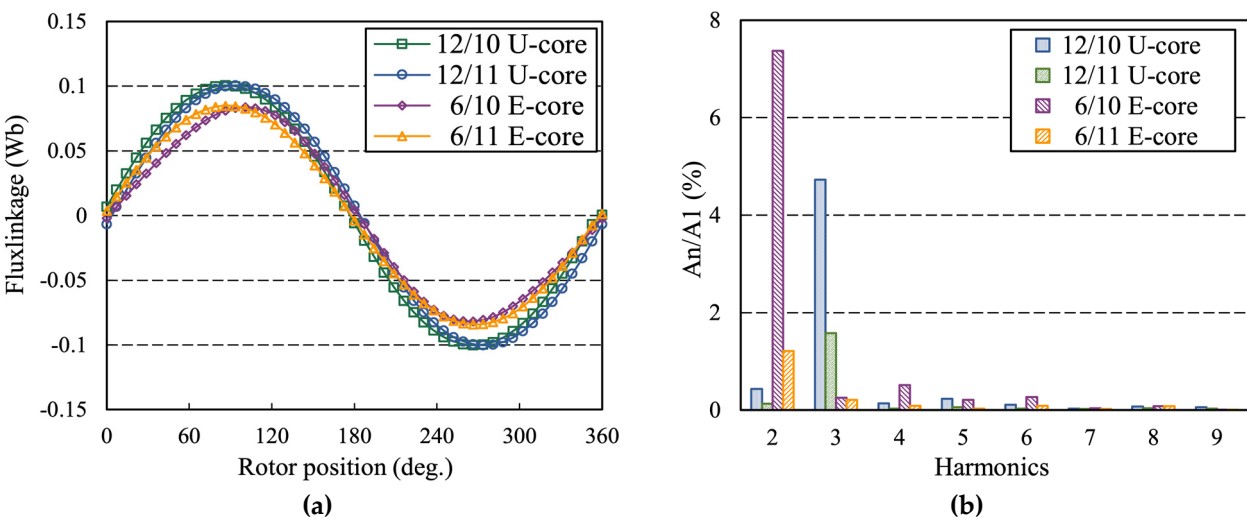

**Figure 5.** No-load air-gap PM flux linkage and harmonic distribution of the four candidates. (**a**) PM flux linkage waveforms; (**b**) harmonic distributions.

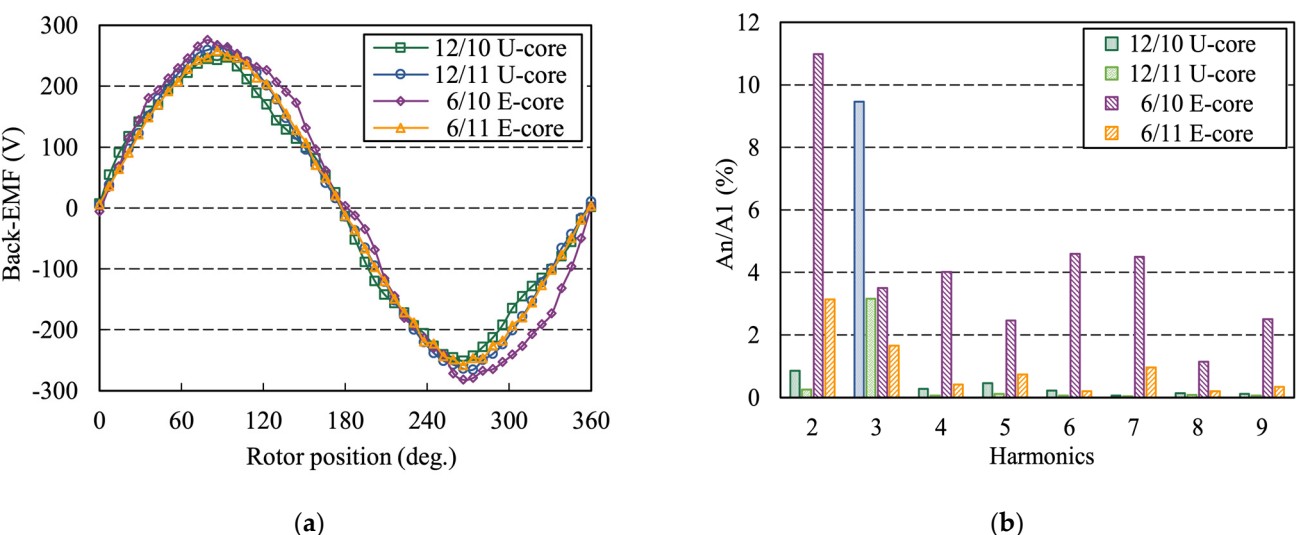

**Figure 6.** No-load phase back EMF and harmonic distribution of the four candidates. (**a**) Phase back EMF; (**b**) harmonic distributions.

### 4.3. Inductance

The influence of both the permanent magnet field and armature magnetic field on the inductance need to be taken into consideration. The self-inductance and mutual inductance of three phase armature windings are defined as follows (take phase A as an example):

$$L_{aa} = (\psi_a - \psi_{pma})/i_a \tag{3}$$

$$M_{an} = (\psi_a - \psi_{pma})/i_n (n = b \ or \ c) \tag{4}$$

where $L_{aa}$ is the self-inductance of phase A; $M_{an}$ is the mutual inductance between phase A and other phases (phase B and phase C); $\psi_{pma}$ is the PM flux linkage of phase A; $\psi_a$ is the total flux-linkage of phase A; $i_a$ is the current of phase A; and $i_n$ is the current of phase B or C.

The self-inductance and mutual inductance of the four candidates are given in Figure 7. It can be seen that the waveforms of three-phase self-inductance and mutual inductance are approximately sinusoidal. Both the self-inductance and mutual-inductance of the 11-rotor pole-pairs machines are larger than that of the 10 rotor pole-pairs machines. Due to the single layer winding of the E-core machines, they exhibit less mutual-inductance and larger self-inductance than U-core AFFSPM machines, which indicates a better fault-tolerant operation capability. The smaller mutual inductance symbolizes the good magnetic isolation ability between each phase, which will not cause terrible interference to the remaining phases when a phase fails. Furthermore, the concentrated windings lead to a high self-inductance of each phase, which gives an effective limitation of fault current when the winding short-circuit appears.

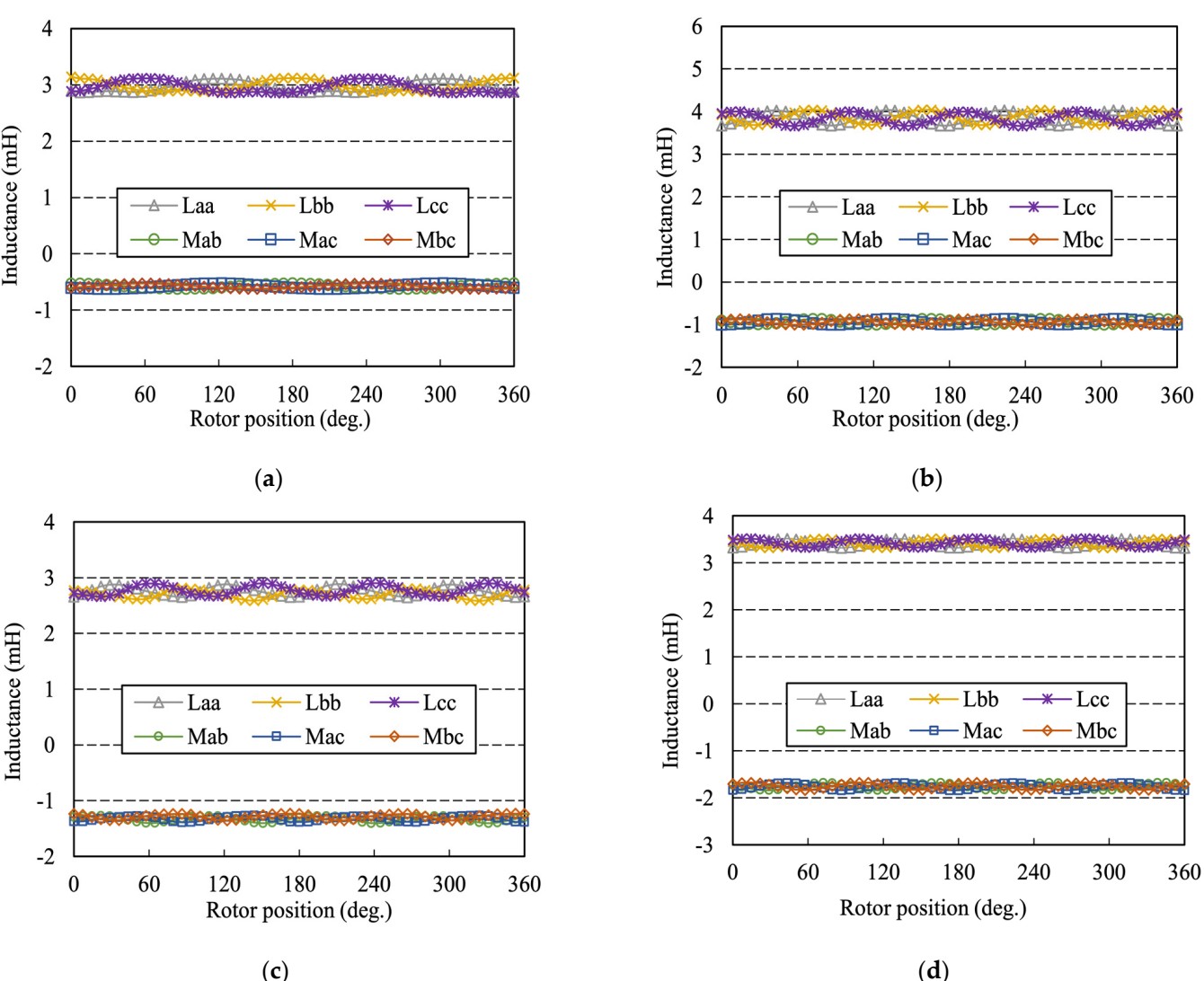

**Figure 7.** Self-inductance and mutual inductance of the four candidates. (**a**) 12/10 U-core; (**b**) 12/11 U-core; (**c**) 6/10 E-core; (**d**) 6/11 E-core.

### 4.4. Electromagnetic Torque

According to the magnetic field modulation principle, the electromagnetic torque of the machine is produced by both primitive harmonics of PM field and armature reaction field, and by the corresponding modulated harmonic components. Due to the negligible difference between d-axis inductance ($L_d$) and q-axis inductance ($L_q$), the reluctance torque can be ignored and $i_d = 0$ control strategies are suitable for AFFSPM machines. Therefore, the electromagnetic torque can be obtained by Equation (5) [24].

$$T_e = \frac{P_e}{\omega_r} = \frac{m}{2} N_{ph} P_r \Phi_{\mathrm{PMm}} I_{am} \tag{5}$$

where $P_e$ is the electromagnetic power, $m$ is the phase number, $N_{ph}$ is the winding turns per phase, $I_{am}$ is the amplitude of the fundamental phase armature current. The electromagnetic torque $T_e$ of the four candidates is shown in Figure 8. The torque ripple is mainly contributed by the cogging torque, which should be reduced by further optimization. Therefore, it can be seen from Figure 8 that the torque ripple of the 12/10 U-core AFFSPM machine is the largest, 17.64%, and the 6/11 E-core AFFSPM machine has the smallest torque ripple, 8.29%.

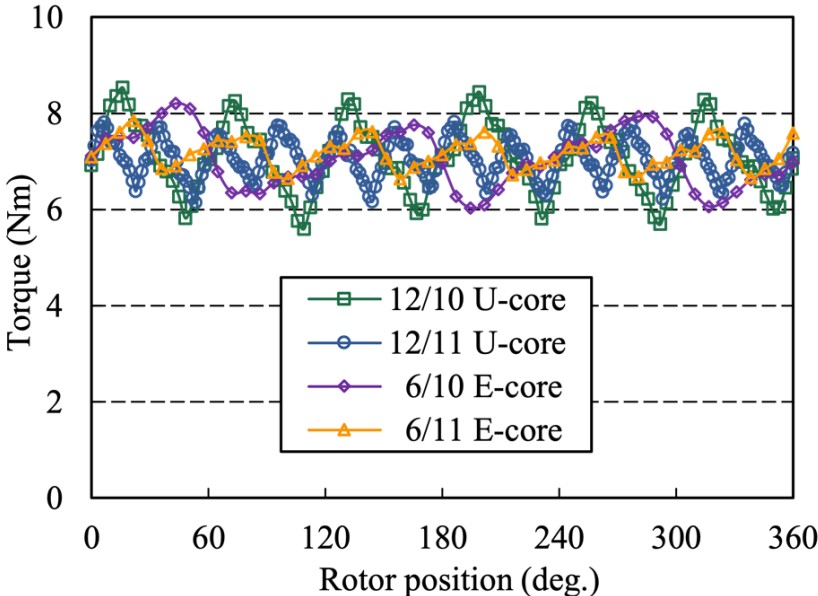

**Figure 8.** Electromagnetic torque of the four candidates.

## 5. Fault-Tolerant Operation Performance Comparison

The fault-tolerant operating performance of the four candidates is analyzed and compared by the 3D FEA method when open-circuit faults occur in one- and two-phase windings. As given in Figure 3, due to the unique structure of AFFSPM machines, two sets of three phases armature windings are arranged into two stators, respectively, and the natural physical isolation between two set windings can give a certain improvement on the fault-tolerant capability.

During the healthy operation where all coils work, the three-phase armature currents and the total magnetomotive force are given:

$$\begin{cases} i_{A1} = i_{A2} = I_m \cdot \sin(\omega t) \\ i_{B1} = i_{B2} = I_m \cdot \sin(\omega t - 2\pi/3) \\ i_{C1} = i_{C2} = I_m \cdot \sin(\omega t - 4\pi/3) \end{cases} \tag{6}$$

$$
\begin{aligned}
f_{total} &= (f_{A1} + f_{B1} + f_{C1}) + (f_{A2} + f_{B2} + f_{C2}) \\
&= 2 * (F_m \cdot \sin(\omega t) \cdot \sin x + F_m \cdot \sin(\omega t - \tfrac{2\pi}{3}) \cdot \sin(x - \tfrac{2\pi}{3}) + F_m \cdot \sin(\omega t - \tfrac{4\pi}{3}) \cdot \sin(x - \tfrac{4\pi}{3})) \\
&= 3F_m \cos(\omega t - x)
\end{aligned}
\tag{7}
$$

where $\omega$ is electrical frequency, $I_m$ is the amplitude of the phase current, $x$ is the electric angle position around the airgap, $F_m$ is the amplitude of single phase fundamental magnetmotive force.

Generally, for the fault-tolerant control strategy, larger armature current is required to compensate the MMF produced by the lost phase(s), which has critical requirement on the power level and cost of the converters. However, the converter power level of the AFFSPM machine is limited for low-cost consideration. Therefore, the fault-tolerant control strategy presented in this section makes the amplitude of the armature current limit to the rated value, and a circular rotating field is achieved by regulating the phase angle of the armature currents in the remaining phases.

### 5.1. One-Phase Open

As can be seen in Figure 9, phase A1 is the only open-circuited phase, which is intentional for testing the one-phase open faulty mode. For the fault-tolerant control strategy targeting the unchanged rotating MMF, a circular rotating field can be achieved by regulating the phase angle of the armature currents of the remaining phases. For the remaining phases, the three symmetric phases (A2, B2, C2) can produce a circular rotating magnetic field and contribute half output torque with initial current phase angle. In addition, the two asymmetric phases (B1, C1) need to be regulated to reform a circular rotating magnetic field.

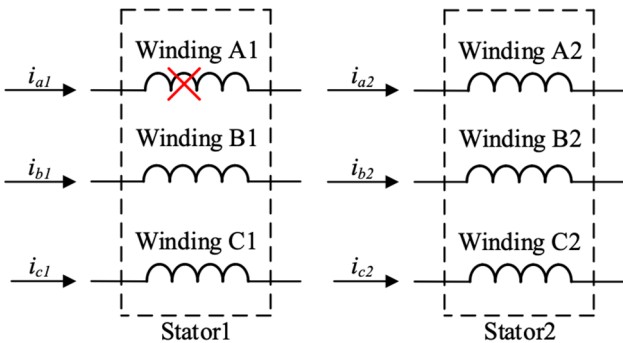

**Figure 9.** Schematic diagram of one-phase open-circuit (phase A1 open-circuit) fault condition.

Suppose that the regulated phase angle of phase B1 and C1 are $\alpha_1$ and $\alpha_2$, respectively. Since the space position of two phases are unchanged, the regulated magnetomotive force can be expressed as Equation (8). In order to obtain a circular rotation magnetic field, the regulated MMF should have the same component as the magnetomotive force given in Equation (7), of which the component except $\cos(\omega t-x)$ in Equation (8) should be eliminated. Hence, the armature current of the remaining phases and the total MMF in one-phase open condition are shown in Equations (9) and (10). The phasor diagram of the regulated current under one-phase open-circuit fault tolerant control is given in Figure 10.

The electromagnetic torque under one-phase open and fault-tolerant control are presented in Figure 11. The average torque between healthy and fault-tolerant condition are compared in Figure 12 and the torque ripple is also noted. It can be found that the E-core AFFSPM machines perform lower torque ripple than U-core AFFSPM machines due to the smaller mutual-inductance of single layer winding. The average torque under the fault-tolerant control is about 77.36% of healthy torque, which is in accordance with the total MMF analysis mentioned above. The 12/10 U-core AFFSPM machine has the most

undesirable torque ripple, and the 6/11 E-core AFFSPM machine exhibits the best torque performance, 5.37 N·m with 9.54% torque ripple
.

$$
\begin{aligned}
f_{total\_regulated} &= f_{B1\_regulated} + f_{C1\_regulated} \\
&= F_m \cdot \sin(\omega t - \alpha_1) \cdot \sin(x - \tfrac{2\pi}{3}) + F_m \cdot \sin(\omega t - \alpha_2) \cdot \sin(x + \tfrac{2\pi}{3}) \\
&= \tfrac{F_m}{2} \cdot \Big[ \cos(\omega t - x) \cdot (-\tfrac{1}{2}\cos\alpha_1 + \tfrac{\sqrt{3}}{2}\sin\alpha_1) + \sin(\omega t - x) \cdot (-\tfrac{1}{2}\sin\alpha_1 - \tfrac{\sqrt{3}}{2}\cos\alpha_1) \\
&\quad - \cos(\omega t + x) \cdot (-\tfrac{1}{2}\cos\alpha_1 - \tfrac{\sqrt{3}}{2}\sin\alpha_1) - \sin(\omega t + x) \cdot (-\tfrac{1}{2}\sin\alpha_1 + \tfrac{\sqrt{3}}{2}\cos\alpha_1) \Big] \\
&\quad + \tfrac{F_m}{2} \cdot \Big[ \cos(\omega t - x) \cdot (-\tfrac{1}{2}\cos\alpha_2 - \tfrac{\sqrt{3}}{2}\sin\alpha_2) + \sin(\omega t - x) \cdot (-\tfrac{1}{2}\sin\alpha_2 + \tfrac{\sqrt{3}}{2}\cos\alpha_2) \\
&\quad - \cos(\omega t + x) \cdot (-\tfrac{1}{2}\cos\alpha_2 + \tfrac{\sqrt{3}}{2}\sin\alpha_2) - \sin(\omega t + x) \cdot (-\tfrac{1}{2}\sin\alpha_2 - \tfrac{\sqrt{3}}{2}\cos\alpha_2) \Big]
\end{aligned}
\tag{8}
$$

$$
\begin{cases}
i_{A2} = I_m \cdot \sin(\omega t) \\
i_{B1} = I_m \cdot \sin(\omega t - 5\pi/6) \\
i_{B2} = I_m \cdot \sin(\omega t - 2\pi/3) \\
i_{C1} = I_m \cdot \sin(\omega t + 5\pi/6) \\
i_{C2} = I_m \cdot \sin(\omega t - 4\pi/3)
\end{cases}
\tag{9}
$$

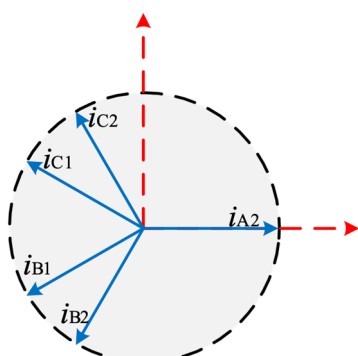

**Figure 10.** Phase diagram of regulated currents under one-phase open-circuit (phase A1 open-circuit) fault-tolerant control.

$$
\begin{aligned}
f_{total\_onephaseopen} &= f_{A2} + f_{B1} + f_{B2} + f_{C1} + f_{C2} \\
&= F_m \cdot \sin(\omega t) \cdot \sin x + F_m \cdot \sin(\omega t - \tfrac{5\pi}{6}) \cdot \sin(x - \tfrac{2\pi}{3}) + F_m \cdot \sin(\omega t + \tfrac{5\pi}{6}) \cdot \sin(x - \tfrac{2\pi}{3}) \\
&\quad + F_m \cdot \sin(\omega t - \tfrac{4\pi}{3}) \cdot \sin(x - \tfrac{4\pi}{3}) + F_m \cdot \sin(\omega t - \tfrac{4\pi}{3}) \cdot \sin(x - \tfrac{4\pi}{3}) \\
&= \tfrac{3+\sqrt{3}}{2} F_m \cos(\omega t - x)
\end{aligned}
\tag{10}
$$

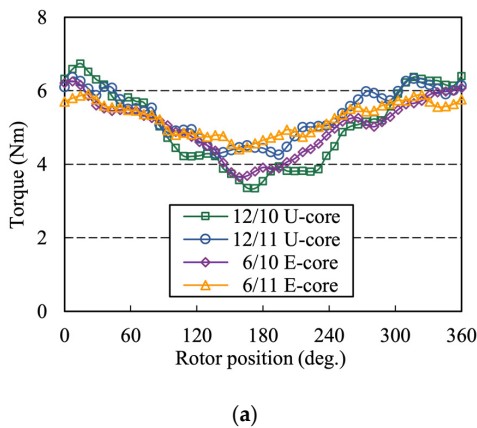
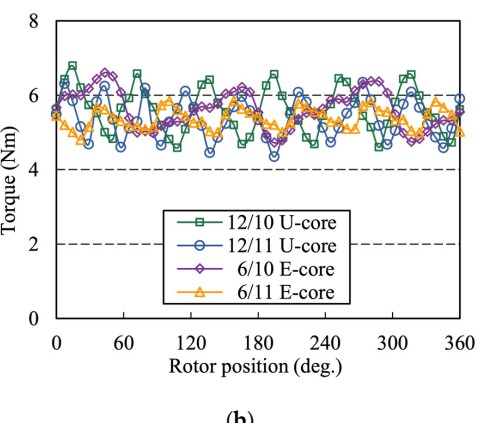

(a)      (b)

**Figure 11.** Electromagnetic torque under one-phase open-circuit and fault-tolerant control operation condition of four candidates. (**a**) One-phase open-circuit; (**b**) fault-tolerant control operation.

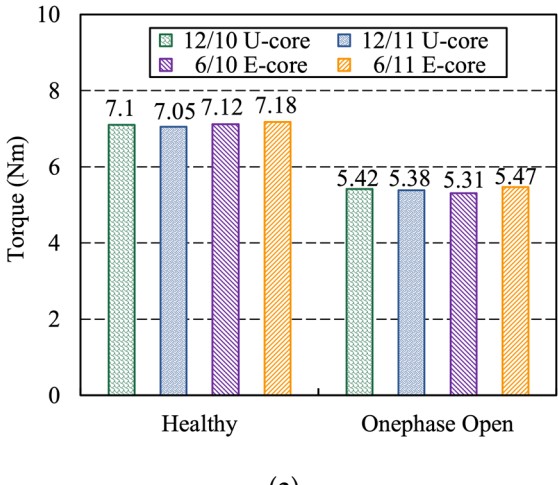
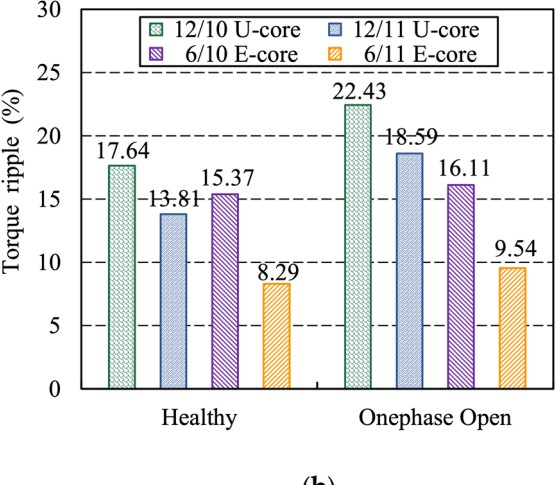

(**a**)                                    (**b**)

**Figure 12.** Comparison of electromagnetic torque and torque ripple of four candidates. between healthy and one-phase open fault-tolerant control operation. (**a**) Average electromagnetic torque; (**b**) Torque ripple.

### 5.2. Two-Phase Open

There are three cases for two-phase-open faults. Case I: same two phases in two sets (A1 and A2 open); case II: two different phases in two sets (A1 and B2 open); case III: two different phases in one set (A1 and B1 open), which is sketched in Figure 13. Similar to the fault-tolerant strategy for one-phase open-circuit fault, the remaining four phases under three cases of two phases open-circuit can be divided into two-phase groups. Thus, the current of each group can be regulated to form a circular rotating magnetic field without any additional conditions. The regulated current of remaining phases corresponding to each case are given to achieve the maximum output torque in fault-tolerant operation, by the same method applied in one-phase open condition.

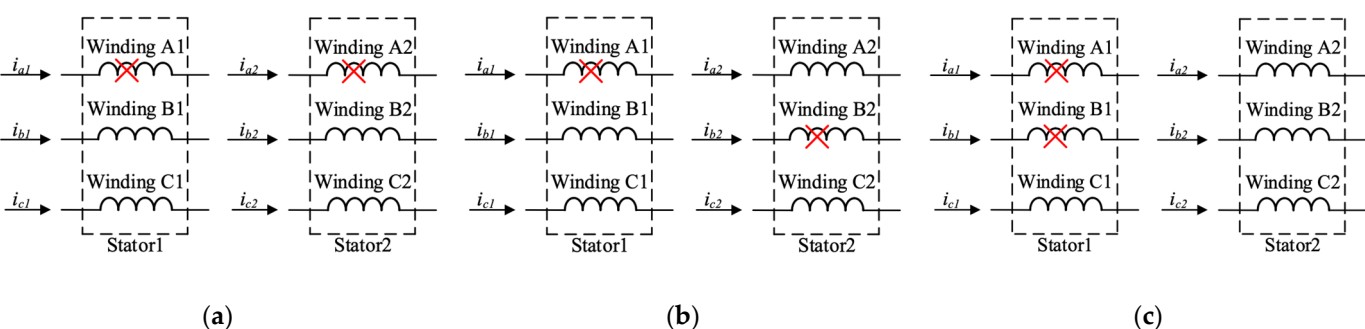

(**a**)                             (**b**)                             (**c**)

**Figure 13.** Winding configurations of two-phases open-circuited under three fault cases. (**a**) Case I; (**b**) Case II; (**c**) Case III.

In case I, the two groups are phase B1, C1 and phase B2, C2, respectively. The regulated current and total magnetomotive force are given as Equations (14) and (15), respectively. And the phasor diagram of the regulated current under Case I is given in the Figure 14 in addition.

$$
\begin{aligned}
group\ 1 &: \begin{cases} i_{B1} = I_m \cdot \sin(\omega t - 5\pi/6) \\ i_{C1} = I_m \cdot \sin(\omega t + 5\pi/6) \end{cases} \\
group\ 2 &: \begin{cases} i_{B2} = I_m \cdot \sin(\omega t - 5\pi/6) \\ i_{C2} = I_m \cdot \sin(\omega t + 5\pi/6) \end{cases}
\end{aligned}
\tag{11}
$$

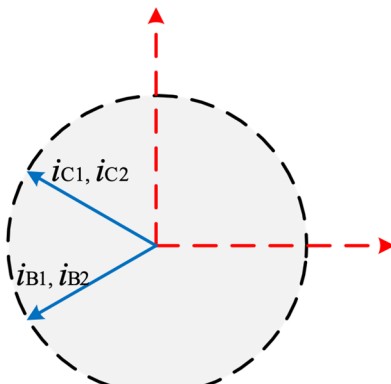

**Figure 14.** Phase diagram of regulated currents under two-phase open-circuit (Case I: phase A1 and A2 open-circuit) fault-tolerant control.

$$
\begin{aligned}
f_{total\_caseI} &= f_{B1} + f_{B2} + f_{C1} + f_{C2} \\
&= F_m \cdot \sin(\omega t - \tfrac{5\pi}{6}) \cdot \sin x + F_m \cdot \sin(\omega t - \tfrac{5\pi}{6}) \cdot \sin(x - \tfrac{2\pi}{3}) \\
&\quad + F_m \cdot \sin(\omega t + \tfrac{5\pi}{6}) \cdot \sin(x - \tfrac{4\pi}{3}) + F_m \cdot \sin(\omega t + \tfrac{5\pi}{6}) \cdot \sin(x - \tfrac{4\pi}{3}) \\
&= \sqrt{3} F_m \cos(\omega t - x)
\end{aligned} \tag{12}
$$

In case II, the two groups are phase B1, C1 and phase A2, C2, respectively. The expression and phase diagram of the regulated current are given in Equation (13) and Figure 15, respectively. Moreover, the total magnetomotive force are expressed in Equation (14).

$$
\begin{aligned}
group\ 1: &\begin{cases} i_{B1} = I_m \cdot \sin(\omega t + 5\pi/6) \\ i_{C1} = I_m \cdot \sin(\omega t - 5\pi/6) \end{cases} \\
group\ 2: &\begin{cases} i_{A2} = I_m \cdot \sin(\omega t - \pi/6) \\ i_{C2} = I_m \cdot \sin(\omega t + \pi/2) \end{cases}
\end{aligned} \tag{13}
$$

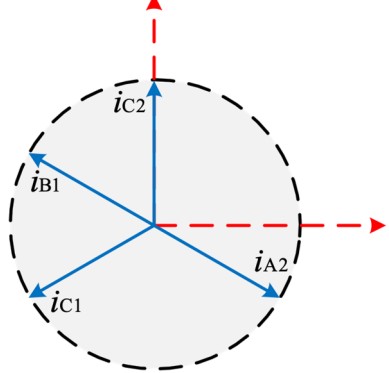

**Figure 15.** Phase diagram of regulated currents under two-phase open-circuit (Case II: phase A1 and B2 open-circuit) fault-tolerant control.

$$
\begin{aligned}
f_{total\_caseII} &= f_{A2} + f_{B1} + f_{C1} + f_{C2} \\
&= F_m \cdot \sin(\omega t - \tfrac{\pi}{6}) \cdot \sin x + F_m \cdot \sin(\omega t + \tfrac{5\pi}{6}) \cdot \sin(x - \tfrac{2\pi}{3}) \\
&\quad + F_m \cdot \sin(\omega t - \tfrac{5\pi}{6}) \cdot \sin(x - \tfrac{4\pi}{3}) + F_m \cdot \sin(\omega t + \tfrac{\pi}{2}) \cdot \sin(x - \tfrac{4\pi}{3}) \\
&= \sqrt{3} F_m \cos(\omega t - x)
\end{aligned} \tag{14}
$$

In case III, the two groups are phase A2, C1 and phase B2, C2, respectively. The regulated current and total magnetomotive force are presented in Equations (15) and (16). The phase diagram of two groups current is shown in Figure 16.

$$
\begin{aligned}
group\ 1: &\begin{cases} i_{A2} = I_m \cdot \sin(\omega t - \pi/6) \\ i_{C1} = I_m \cdot \sin(\omega t + \pi/2) \end{cases} \\
group\ 2: &\begin{cases} i_{B2} = I_m \cdot \sin(\omega t + 5\pi/6) \\ i_{C2} = I_m \cdot \sin(\omega t - 5\pi/6) \end{cases}
\end{aligned}
\tag{15}
$$

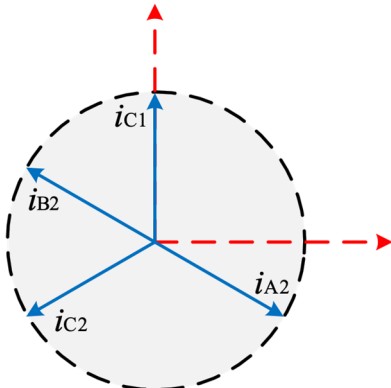

**Figure 16.** Phase diagram of regulated currents under two-phase open-circuit (Case III: phase A1 and B1 open-circuit) fault-tolerant control.

$$
\begin{aligned}
f_{total\_caseIII} &= f_{A2} + f_{B2} + f_{C1} + f_{C2} \\
&= F_m \cdot \sin(\omega t - \tfrac{\pi}{6}) \cdot \sin x + F_m \cdot \sin(\omega t + \tfrac{5\pi}{6}) \cdot \sin(x - \tfrac{2\pi}{3}) \\
&\quad + F_m \cdot \sin(\omega t - \tfrac{5\pi}{6}) \cdot \sin(x - \tfrac{4\pi}{3}) + F_m \cdot \sin(\omega t + \tfrac{\pi}{2}) \cdot \sin(x - \tfrac{4\pi}{3}) \\
&= \sqrt{3} F_m \cos(\omega t - x)
\end{aligned}
\tag{16}
$$

The torque performance of four candidates under different fault conditions and corresponding fault-tolerant control is obtained based on the 3-D FEA method, which is outlined in Figures 17–19, respectively. To make a clear comparison, the average torque between the healthy and fault-tolerant of three cases are compared in Figure 20, in which the torque ripple is also recorded.

Although the same circle rotation magnetic field is obtained under the fault-tolerant control in three cases, the relative positions of the fault-phase result in the varieties in the torque ripple. Case II has the least torque ripple and Case III accounts for the largest torque ripple. However, the U-core machines demonstrated a larger torque ripple than the E-core machines in all three cases.

From the view of the machine topologies and winding configurations, some explanations can be drawn. In Case I, the same phases in two sets open-circuited can be regarded as a double superposition of one-phase open conditions and results in a larger torque ripple than the situation analyzed in Section 5.1. There is also one-phase open-circuited in each set winding in Case II, but the fault-phase is not the same, which makes the torque ripple of each group partially offset and produces the smoothest waveform of total electromagnetic torque among the three cases. As for Case III, since two failed phases come from one set, the remaining one phase needs to be grouped with one phase from another set of windings. Such a winding group arrangement will lead to an uneven MMF distribution and undesirable fluctuation in the torque wave, which results in the worst torque performance of all cases. The small mutual-inductance of E-core machines can increase the magnetic isolation between the remaining phases and reduce the impact on the uneven distribution of MMF due to the regulated armature current, so as to make the total MMF as close to the circle as possible and bring a desirable torque performance with the small torque ripple. In

addition, due to the winding complementarity, there are the better performance in 12/11 U-core machine and 6/11 E-core machine than the 12/10 U-core machine and 6/10 E-core machine, respectively.

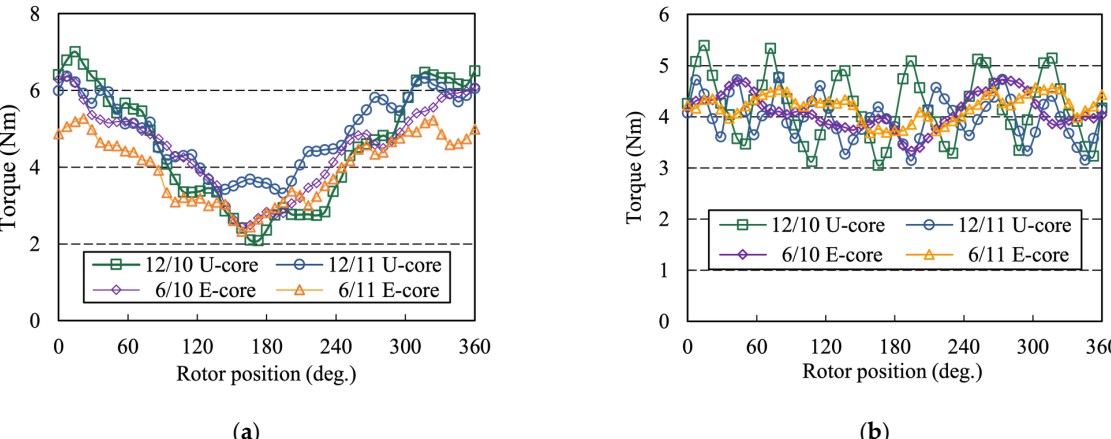

(**a**)                                       (**b**)

**Figure 17.** Electromagnetic torque under case I (phase A1, A2 open) and fault-tolerant control condition of four candidates. (**a**) Case I (phase A1, A2 open-circuit); (**b**) fault-tolerant control.

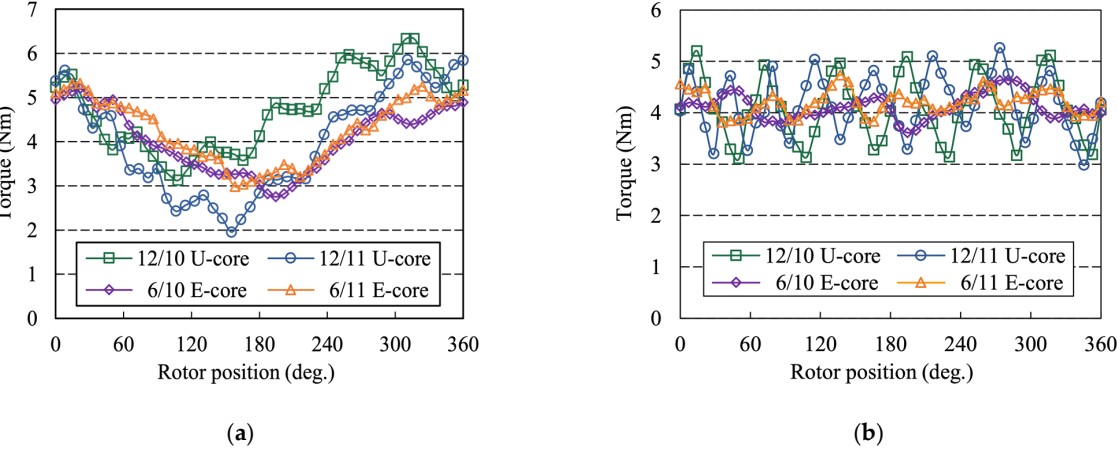

(**a**)                                       (**b**)

**Figure 18.** Electromagnetic torque under case II (phase A1, B2 open) and fault-tolerant control condition of four candidates. (**a**) Case II (phase A1, B2 open-circuit); (**b**) fault-tolerant control.

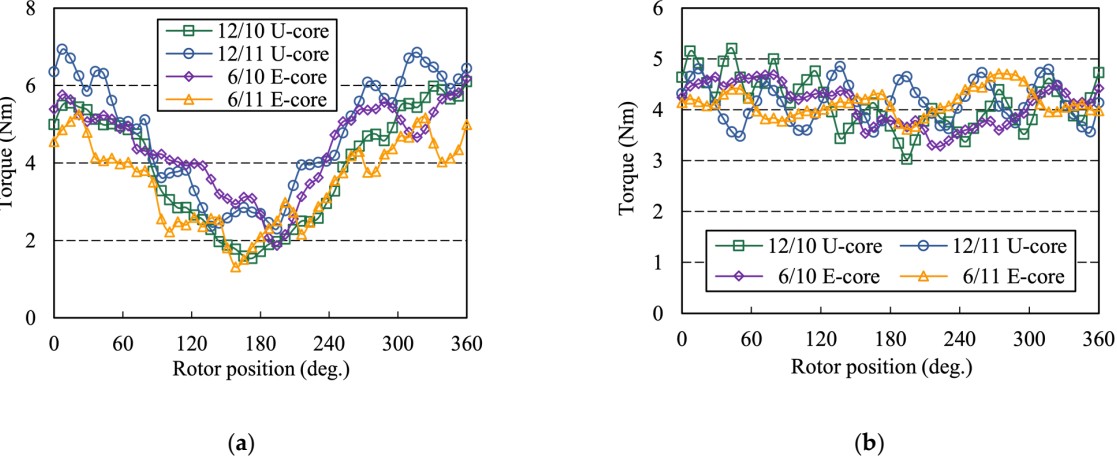

(**a**)                                       (**b**)

**Figure 19.** Electromagnetic torque under case III (phase A1, B1 open) and fault-tolerant control condition of four candidates. (**a**) Case III (phase A1, B1 open-circuit); (**b**) fault-tolerant control.

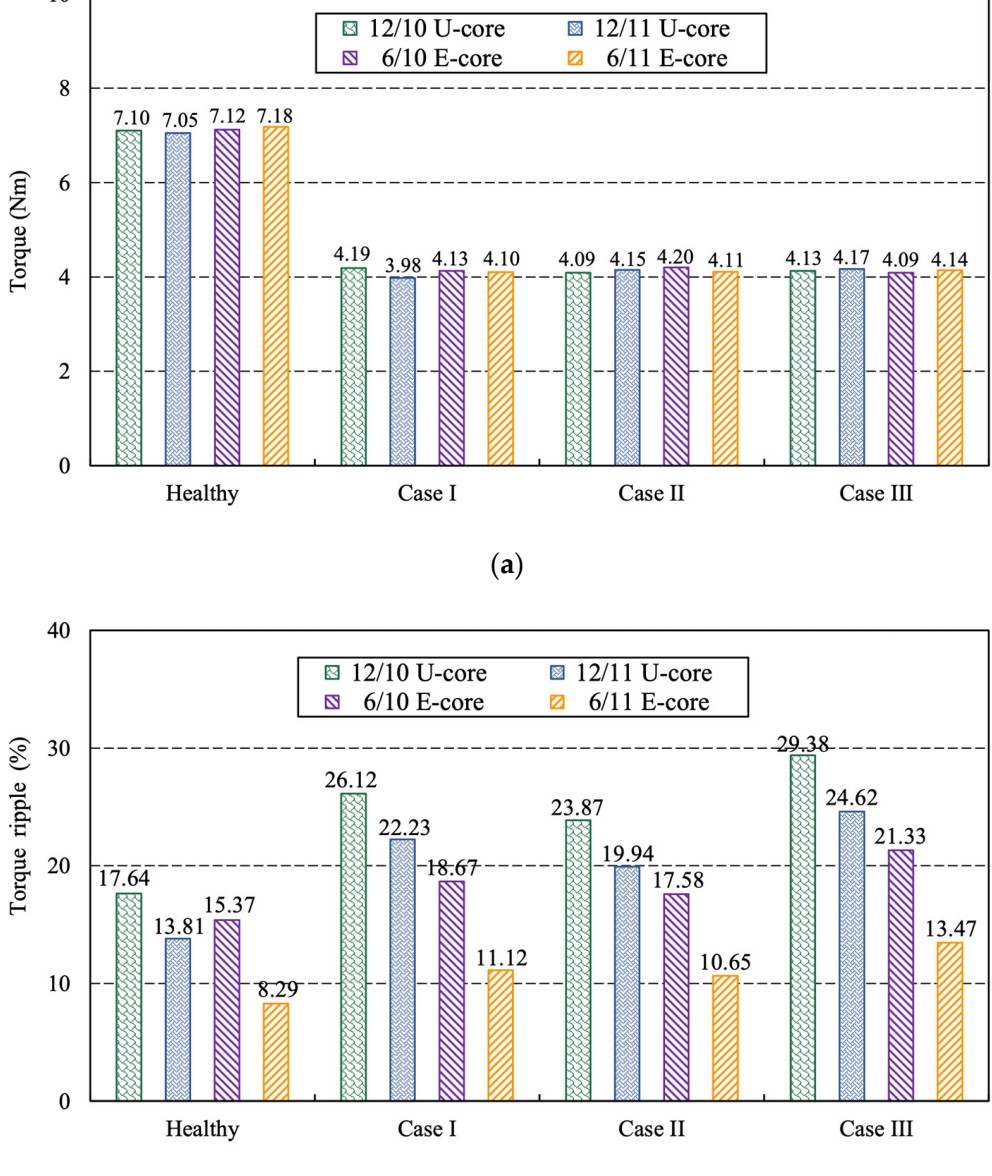

**Figure 20.** Comparison of electromagnetic torque and torque ripple of four candidates between healthy and two-phase open-circuited fault-tolerant control operation. (**a**) Average electromagnetic torque; (**b**) torque ripple.

## 6. Conclusions

In this paper, AFFSPM machines with U-core and E-core stator modular segments, as well as different slot/pole-pairs combinations, are compared comprehensively between four candidates, including the $P_s/P_r$ combination, static characteristics and fault-tolerance capability. Some conclusions can be summarized as follows:

(1) The E- and U-core AFFSPM machines apply different feasible stator-slot/ rotor-pole-pairs combinations, which can be obtained by the slot-conducotor back-EMF star vectors theory.

(2) The total harmonic distortion (THD) value of PM flux linkage and back-EMF of 10 rotor pole-pairs machines is higher than that of the 11-rotor pole-pairs machines because of the winding complementarity. Furthermore, the effective harmonics of open-circuit air-gap flux of four candidates can be classified into the dominant harmonic and modulated harmonics, which is in agreement with the field modulation theory.

(3) Both U- and E-core machines consist of two stators and one rotor. The two sets windings on each stator provide a natural convenience for double three phases configurations, which can enhance the fault-tolerant capability.

(4) The middle stator teeth of E-core stator segments lead to an enhancement of the magnetic isolation and the single layer windings, which significantly improves the fault-tolerant operation ability under one- and two-phase open-circuited conditions.

(5) The E-core stator segments reduce the number of PMs and increase the distance between adjacent PMs, which not only leads to a smaller cogging torque and torque ripple than that of U-core AFFSPM machines, but also provides a lower torque ripple in the fault tolerant control operation.

**Author Contributions:** Conceptualization, Y.T. and Y.K.; Data curation, Y.T.; Funding acquisition, M.L. and D.X.; Investigation, Y.T.; Methodology, Y.T., Y.K. and D.X.; Project administration, M.L. and K.L.; Software, Y.T.; Supervision, Y.K.; Writing—original draft, Y.T.; Writing—review & editing, M.L. and K.L. All authors have read and agreed to the published version of the manuscript.

**Funding:** This research was founded in part by the National Nature Science Foundation of China under Grant 51937002 and 51807093.

**Institutional Review Board Statement:** Not applicable.

**Informed Consent Statement:** Not applicable.

**Data Availability Statement:** Not applicable.

**Conflicts of Interest:** The authors declare no conflict of interest.

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
