# Peer review of "Comprehensive Comparison of Two Fault Tolerant Axial Field Modular Flux-Switching Permanent Magnet Machines with Different Stator and Rotor Pole-Pairs Combinations"

_machines, doi:10.3390/machines10030201_

Round 1

Reviewer 1 Report

The paper proposes the comparison axial field flux switching PM machine with U-core and E-core segments, respectively with fault tolerant capability. The authors must provide some comments about the following issues:

  • It is stated that for fair comparison key geometric parameters are kept the same; however, it is not sure that such condition will be the best for each one. For instance, the angular width of the stator middle teeth can affect the torque ripple, but it is not clear if the value in Table 1 is the best for the chosen design. There are also some other differences of geometric sizes that should be explained much more.
  • Figure 4 shows the air-gap flux density on a stationary angular coordinate; therefore, the use of ‘rotor position’ is misleading. I believe that the rotor position is fixed. If not so, please clarify what condition you have considered. Moreover, I believe the curves come from a finite element analyses, but it is not specified in the text.
  • It is not clear from Figs 5b and 6b why A2/A1 is >7% for the phase flux linkage and ~5.5% for the phase back-emf. I expect that the latter be higher than the former one. May you explain why such a result?

Minor suggestions are:

  • Some figures would be more legible by reducing the samples as they shadow the curve profile
  • Check some typos in section 5
  • I think that ‘capability’ sounds better than ‘capacity’

Author Response

Dear Reviewer:

We greatly appreciate you for your peer inspection and helpful comments to the key parts worth reconsidering. I highly value your recommendations and have tried my best to address the mistakes pointed out by you. You indicated several parts needed to be corrected in the paper. The corrections and the corresponding explanations are given in the attachment.

Reviewer 2 Report

The manuscript is well written and has a logical structure. The studied electrical machines (Axial Field Modular Flux-Switching Permanent Magnet Machines) are widely used electromechanical devices nowadays and fault-tolerant application is very important to the industry. 

The introduction is well written and the research goal is clearly defined. 

The methodology section is describing the studied topologies, with clear illustrations and a detailed description of the faults. 

The research is based on a simulation that seems to be valid. Conclusions are based on simulation results and answer the research questions declared in the introduction section. 

The manuscript might be accepted to MDPI Machines after minor revision - proofreading and formatting the manuscript. The style of references used in the manuscript must be updated according to the template.  Fig. is used together with Figure, etc.

Author Response

(The authors gave the same response as above.)

Reviewer 3 Report

This paper introduces interesting topic and my comments are as follows:

1- The English language must be modified  for example in page 11 line 282 ,"ror" should be"for".

2- in the title of the paper you said you will compare two fault tolerant, however, in abstract you said four fault tolerant?

3-Do you consider the effects of saturation during fault, it has a significant effect on the fault performance of the machine?

4- describe how you make harmonic analysis for flux density in figure 4 and give the value of the THD?.

5-Equation 13 is better to describe the reconstructed currents using phasor diagram.

6- The recent literature must be discussed in details in the introduction section.

Author Response

(The authors gave the same response as above.)

Round 2

Reviewer 3 Report

The authors have answered my questions and I am satisfied about their answer. Hence, the paper can be accepted in its present form.